# Impact of the Protein Environment on Two-Photon Absorption Cross-Sections of the GFP Chromophore Anion Resolved at the XMCQDPT2 Level of Theory

**DOI:** 10.3390/ijms241411266

**Published:** 2023-07-10

**Authors:** Vladislav R. Aslopovsky, Andrei V. Scherbinin, Nadezhda N. Kleshchina, Anastasia V. Bochenkova

**Affiliations:** Department of Chemistry, Lomonosov Moscow State University, Leninskie Gory 1/3, 119991 Moscow, Russia

**Keywords:** two-photon absorption, green fluorescent protein, multi-state multi-reference perturbation theory, XMCQDPT2

## Abstract

The search for fluorescent proteins with large two-photon absorption (TPA) cross-sections and improved brightness is required for their efficient use in bioimaging. Here, we explored the impact of a single-point mutation close to the anionic form of the GFP chromophore on its TPA activity. We considered the lowest-energy transition of EGFP and its modification EGFP T203I. We focused on a methodology for obtaining reliable TPA cross-sections for mutated proteins, based on conformational sampling using molecular dynamics simulations and a high-level XMCQDPT2-based QM/MM approach. We also studied the numerical convergence of the sum-over-states formalism and provide direct evidence for the applicability of the two-level model for calculating TPA cross-sections in EGFP. The calculated values were found to be very sensitive to changes in the permanent dipole moments between the ground and excited states and highly tunable by internal electric field of the protein environment. In the case of the GFP chromophore anion, even a single hydrogen bond was shown to be capable of drastically increasing the TPA cross-section. Such high tunability of the nonlinear photophysical properties of the chromophore anions can be used for the rational design of brighter fluorescent proteins for bioimaging using two-photon laser scanning microscopy.

## 1. Introduction

The two-photon excitation of fluorescence was first applied to laser scanning microscopy in 1990 [1]. Since then, two-photon laser scanning microscopy (2PLSM) has been actively developed for high-resolution imaging in living tissues. This approach provides several advantages compared to single-photon confocal microscopy, such as higher contrast and resolution, an increased penetration depth of near-infrared radiation, and lower phototoxicity [2,3]. However, genetically encoded fluorescent proteins (FPs) commonly used in bioimaging were not originally optimized for applications in 2PLSM, and therefore, they do not have optimal two-photon absorption (TPA) properties, such as TPA cross-sections and brightness. The latter is defined as the TPA cross-section multiplied by the fluorescence quantum yield. The search for fluorescent proteins with larger TPA cross-sections and, hence, improved brightness is required to minimize the heat-induced damage of tissues by reducing the intensities of excitation laser pulses [4].

In fluorescent microscopy, enhanced green fluorescent protein (EGFP) [5] is one of the most-commonly used engineered variants of the original wild-type green fluorescent protein [6,7]. The GFP chromophore is 4-hydroxybenzylidene-2,3-dimethylimidazolinone (HBDI), which is formed upon the spontaneous cyclization and oxidation of the sequence -Ser65 (or Thr65)-Tyr66-Gly67-. Two-point mutations that generate EGFP, F64L, and S65T (here and below, substitutions from the primary sequence are given as the single-letter code for the amino acid being replaced, its numerical position in the sequence, and the single-letter code for the replacement) contribute to its improved properties. EGFP has greater folding efficiency at 37 °C due to the F64L mutation. The S65T mutation is responsible for a single excitation peak of EGFP at ∼490 nm, which is in contrast to wild-type GFP with two separate excitation peaks observed due to the coexistence of neutral, protonated ($\lambda_{ex}∼$395 nm) and anionic, deprotonated ($\lambda_{ex}∼$490 nm) forms of the chromophore [8]. In EGFP, the excitation peak at 395 nm is suppressed by S65T due to the modulation of a charged state of Glu222 involved in the hydrogen bonding network close to the chromophore (see Figure 1). The Glu222 residue in turn defines a charged state of the chromophore, which becomes stabilized as the anion in the ground state when Glu222 is protonated [9]. The deprotonated form of the chromophore is further stabilized by the hydrogen bonding interactions with His148, Thr203, and a conserved water molecule. The anionic form of the GFP chromophore is responsible for the fluorescence of the protein, and EGFP is characterized by increased brightness upon one-photon excitation as compared to wild-type GFP [10].

A colorful palette of fluorescent proteins with different absorption/emission spectra has been discovered in various organisms [12] and has also been engineered by means of direct mutagenesis [13]. Single-point mutations enable re-engineering the FP proteins to improve their spectral properties for particular applications. Indeed, the experimental TPA properties of FP mutants have been reported [14], and the two-photon directed evolution of EGFP has resulted in 50% enhancement of the TPA cross-section and brightness at the peak position [15]. The corresponding structure–property relationship for a series of FPs with the same chromophore in different local protein environments has been proposed [16,17]. The internal electric field of the protein, which can be altered by introducing point mutations, is shown to play a major role in the TPA properties of EGFP by modulating the changes of the permanent dipole moments of the chromophore in the excited and ground electronic states [4].

Theoretical simulations of TPA cross-sections and excitation wavelengths are required to provide a quantitative relation between the TPA properties and the protein structure. There are two main groups of approaches for calculating TPA matrix elements. The so-called “sum over states” approaches go back to time-dependent perturbation theory, which represents the TPA matrix elements through an infinite series over intermediate states. In practical applications, it is approximated by a finite sum. Apart from the questionable truncation procedure, this approach requires reliable estimates of the excited state properties, such as transition energies and dipole moment matrix elements. In the alternative group of approaches, these problems are avoided either by means of quadratic response theory, in which the TPA matrix elements are determined using the second-order response function for a system exposed to external time-dependent perturbation [18], or using a somewhat different “expectation-value” formalism [19]. Approaches from the second group are currently implemented for a variety of quantum chemistry methods, including the time-dependent Hartree–Fock [20,21], multiconfiguration self-consistent field (MCSCF) [18,22], coupled-cluster [23,24,25,26], and time-dependent density functional theory (TDDFT) [27,28,29,30,31], as well as the algebraic-diagrammatic construction polarization propagator (ADC) [32] and the equation-of-motion coupled-cluster (EOM-CC) [19,33] methods. Some of these methods have been previously applied to the study of TPA of FPs and their chromophores [19,34,35,36,37].

The sum-over-states approach can in principle be applied to any quantum chemistry method for which the excited state properties are available. For example, it has successfully been applied to calculate the TPA spectra of bovine rhodopsin using the second-order extended multiconfiguration quasidegenerate perturbation theory (XMCQDPT2) [38]. The sum-over-states approach is also very convenient for the interpretation of the TPA process in terms of the combination and interference of various excitation channels, as well as for studying the factors affecting the TPA activity. Importantly, the initial (*i*) and final (*f*) states can also serve as intermediate states for the TPA transition: i→i→f and i→f→f, as theoretically predicted many years ago [39]. Moreover, these excitation channels can make a dominant contribution to two-photon absorption in some molecules that undergo a large change in the permanent dipole moment upon excitation, which was later confirmed experimentally [40]. This observation has led to a very simple approximation called the “two-level model” (or “two-state model”), which has been proven to be very useful for the qualitative consideration of the lowest $S_{0}\to S_{1}$ TPA transition. Such a transition is also supposed to be strongly allowed as a one-photon absorption process ($\mu_{10}\neq 0$). This is indeed true for certain polar chromophores, for example the chromophores of various FPs [4,14,16,17,41,42,43] and some others [44,45,46]. Three-, four-, and generalized few-state models have also been considered [47,48,49,50,51,52,53,54,55].

In the present work, we explored the impact of a single-point mutation in close proximity to the anionic form of the GFP chromophore on the TPA activity of the protein. We considered the lowest $S_{0}\;\to \;S_{1}$ transition of EGFP and its modification T203I EGFP, which significantly alters the hydrogen bonding network close to the phenolate moiety of the chromophore, and also compared the results with those obtained for the isolated chromophore. We focused on a methodology for obtaining reliable TPA cross-sections for mutated proteins, based on conformational sampling using molecular dynamics (MD) simulations and a high-level XMCQDPT2-based quantum mechanical/molecular mechanical (QM/MM) approach. We investigated the numerical convergence of the sum-over-states formalism for a series of *N*-level models and explored the applicability of the two-level model for the lowest-lying transition in the EGFP-based proteins. The contribution of the nearby protein environment and effects of conformational sampling were qualitatively evaluated.

## 2. Results and Discussion

### 2.1. Impact of the T203I Mutation and Conformational Sampling

The QM/MM-optimized geometries of the MD-annealed S65T GFP protein and its modified T203I version are shown in Figure 2. Since the linear optical properties of the S65T GFP protein are not altered by the presence of the F64L mutation in EGFP [9], we refer to S65T GFP and its mutated T203I variant as EGFP and EGFP T203I, assuming that the F64L mutation does not alter much the nonlinear photophysical properties either. Furthermore, the calculated TPA cross-sections were compared to the experimental data for EGFP and EGFP T203I. By introducing the T203I point mutation, the hydrogen bonding pattern close to the chromophore drastically changes. The hydrophobic Ile203 residue does not form a hydrogen bond with the phenolate moiety, thus reducing the stabilization of the anion. The total number of hydrogen bonds at the phenolate oxygen, therefore, changes from 3 (Thr203, His148 and H${}_{2}$O) to 2 (His148 and H${}_{2}$O). This reduction in the stabilization of the deprotonated form indeed results in the co-existence of the anionic and neutral forms of the chromophore in the T203I mutant, as observed experimentally for the GFP protein with similar mutations close to the chromophore [56]. We note that, in contrast to T203I EGFP, the T203I GFP protein without the key S65T mutation exhibited only a single excitation peak due to the neutral form of the chromophore [57].

The T203I mutation also red-shifts the absorption of EGFP due to the loss of a hydrogen bond. The shift is as large as *ca.* 1000 cm${}^{-1}$, both in EGFP T203 ($\lambda_{max}=$ 507 nm) compared to EGFP ($\lambda_{max}=$ 488 nm) [15] and in GFP T203I ($\lambda_{max}=$ 501 nm) compared to GFP ($\lambda_{max}=$ 477 nm) [58] for the one-photon absorption of the anionic form of the GFP chromophore. It has been proposed that the extra hydrogen bond, created by Thr203, pulls the electron density to the phenolate oxygen, thus stabilizing the ground electronic state of the anion and increasing the transition energy as compared to EGFP T203I [17,58]. Our calculated vertical excitation energies (VEE) of EGFP (492 nm) and EGFP T203I (510 nm) reproduce well such a red shift, which equals 720 cm${}^{-1}$, as compared to the experimental one of 770 cm${}^{-1}$ [15].

The disruption of the hydrogen bonding network close to the chromophore may also lead to a larger conformational flexibility of amino acid residues in the chromophore binding site in EGFP T203I. To this point, we performed conformational sampling of EGFP T203I using MD simulations. Figure 3 shows that isoleucine is indeed conformationally more flexible than threonine. There was also a lack of one water molecule close to the imidazolinone ring of the chromophore in the annealed and QM/MM-optimized structure of EGFP T30I compared to EGFP (see Figure 2). The water molecules also became mobile, which was due to a nearby hydrophobic residue introduced by the T203I mutation.

Twenty-one structures along the MD trajectory were randomly chosen followed by their QM/MM optimization for the calculations of the TPA cross-sections. The distribution of the calculated TPA cross-sections, $\sigma_{TPA}$, is shown in Figure 4. First, we note that the distribution had one maximum and a large dispersion, strongly indicating that conformational sampling of EGFP T203I must be taken into account when calculating the TPA cross-sections. Second, it is important that the TPA cross-section of the annealed structure lied in the center of this distribution. This showed that the annealing procedure was capable of providing the most-probable/-stable structure of the protein’s active center characterized by the most-likely $\sigma_{TPA}$ value. The calculated TPA cross-section of EGFP T203I at the distribution maximum (15 GM) was also very close to the experimental one (19 GM) [15].

### 2.2. Analysis of the Calculated TPA Cross-Sections

The calculated Δμ, μ, VEEs, and TPA cross-sections ($\sigma_{NLM}$) obtained using the N-level models for the S${}_{0}$→ S${}_{1}$ transition are shown in Table 1. The calculated TPA cross-sections of EGFP and EGFP T203I were also compared to the experimental values [15]. By comparing our results on the HBDI anion with those obtained previously using the TDDFT quadratic response theory [36], we concluded that the latter significantly underestimated the absolute values of the TPA strengths, $\delta_{TPA}$, due to the deficiency in describing the excited state properties. This is in agreement with previous benchmark calculations on the performance of various exchange correlation functionals for predicting TPA strengths [34,50].

The results presented in Table 1 show that the two-level approximation provided about the same values of $\delta_{TPA}$ for the S${}_{0}$ → S${}_{1}$ transition as the higher *N*-level models. This is consistent with the results obtained previously [16,17], where it was shown that the contribution of the intermediate states, which were higher in energy than $S_{1}$, to the $\sigma_{TPA}$ value was not larger than 5%. This is due to the nature of the S${}_{0}$ → S${}_{1}$ transition in fluorescent proteins, which are characterized by both the large dipole moment transition and the relatively large difference of the permanent dipole moments in the ground and first excited state. In such cases, the contributions from the initial and final states to $\sigma_{TPA}$ become the largest.

For the S${}_{0}$ → S${}_{1}$ transition, the $\sigma_{TPA}$ (and $\sigma_{TLM}$) values increased symbatically with the Δμ values in a row of EGFP T203I → HBDI → EGFP, while the μ values did not change much. Therefore, $\sigma_{TLM}$ mainly depended on $|\Delta \mu_{10}|$ in this case (see Equation (Equation 11)), in contrast to the one-photon absorption process. The calculated TPA cross-sections were in good agreement with the experimental ones for both EGFP and EGFP T203I.

The T203I mutation decreased $|\Delta \mu_{10}|$ and, hence, $\sigma_{TPA}$ for the S${}_{0}$ → S${}_{1}$ transition. This is consistent with the structure–property relationship proposed previously [4]. The internal electric field created by the protein environment (E) induced an additional dipole moment on the chromophore in its ground and first excited states:(1)$$\Delta \mu_{10}=\Delta \mu_{10}^{0}+\Delta \mu_{10}^{ind},$$
where $\Delta \mu_{10}^{0}$ is the difference of the permanent dipole moments in the two states for the isolated chromophore, i.e., in the gas phase, and $\Delta \mu_{10}^{ind}$ is the difference between the induced dipole moments. In Figure 5, the directions of the $\Delta \mu_{10}$ vectors are shown for EGFP and EGFP T203I. This vector reflects a partial charge transfer character upon excitation. The electron density moved from phenolate (P) to imidazolinone (I) upon the S${}_{0}$ → S${}_{1}$ transition, and therefore, $\Delta \mu_{10}$ was directed from I to P. Figure 5 also shows differential electron densities, visualizing how the electron density was redistributed upon the S${}_{0}$ → S${}_{1}$ transition.

If the axis x matches the $\Delta \mu_{10}$ direction, then we can rewrite Equation (Equation 1) in the following form:(2)$${\Delta \mu }_{10}={\Delta \mu }_{10}^{0}+\Delta \alpha_{10}E,$$
where $\Delta \alpha_{10}$ is the difference of the polarizabilities along the x direction between the excited and ground states and $\Delta \mu_{10}^{ind}$ and *E* are the projections of the $\Delta \mu_{10}^{ind}$ and E vectors on the x axis [4]. In order to increase the $\Delta \mu_{10}$ value, thereby increasing the TPA cross-section, the internal field should be applied along the P → I direction, opposite the direction of $\Delta \mu_{10}$, since $\Delta \alpha_{10}$ is shown to be negative in the case of the GFP chromophore [4]. Therefore, a positive charge near the phenolate and a negative charge close to the imidazolinone ring should increase $|\Delta \mu_{10}|$ and, hence, $\sigma_{TPA}$.

From the above consideration, we can conclude that the internal electric field in EGFP T203I along the P → I direction is negligible and does not induce any appreciable dipole moment on the chromophore, resulting in $\sigma_{NLM}$ being similar to that calculated in the gas phase (see Table 1) and much smaller compared to EGFP. It is worth noting that such a drastic change can be achieved by a single mutation, which merely reduces a number of hydrogen bonds to the phenolate oxygen atom. This can be traced to the anionic form of the GFP chromophore, which is easily stabilized even by a single hydrogen bond. The ground state both stabilized and polarized by polar residue Thr203 through a hydrogen bond in EGFP resulted in a blue shift of the S${}_{0}$ → S${}_{1}$ absorption, as well as in larger $|\Delta \mu_{10}|$ upon excitation. This indicates that blue-shifted EGFP-like proteins are expected to be brighter upon two-photon transition, as proposed recently [4]. The broad distribution of $\sigma_{TPA}$ for different structures of EGFP T203I presented in Figure 4 can, thus, be explained by a strong sensitivity of the internal electric field to the environment of the GFP chromophore anion.

## 3. Materials and Methods

### 3.1. Molecular Dynamics Simulations

We used the X-ray structure of the S65T GFP protein (PDB ID 1EMA [59]) for constructing a full atomistic model of the protein with the anionic form of the chromophore in the ground electronic state. We note that the core structure and fold of the EGFP protein with two mutations, S65T and F64L, are very similar to those of the wild-type GFP and S65T GFP proteins [9]. The S65T mutation is critical for the hydrogen bonding pattern and the charged states of the chromophore and its nearby residues. However, the F64L mutation, which is located further away from the chromophore, does not alter the optical properties of the protein and serves to improve the packing of hydrophobic residues, reducing their surface exposure. Therefore, we here considered only one key mutation, S65T. The model system was obtained as described in [11]. Briefly, the X-ray structure was first prepared for the molecular dynamics simulations. Following the solvation of the protein by adding water molecules, the model system with a total number of ∼23,000 atoms was initially optimized for 1000 steps. Periodic boundary conditions were applied during the MD simulations in the NVT ensemble at 300 K. The simulations were carried out with an integration step of 1 fs for 2 ns. The model system was then gradually cooled down to 50 K in steps of 50 K during 1.2 ns to obtain a representative minimum on the multidimensional ground state potential energy surface. The final MD geometry was then obtained by performing geometry optimization for an additional 2000 steps. The annealed MD structure was then reduced in size down to 5410 atoms by removing the outer water molecules located farther than 2.9 Å from the protein for the subsequent QM/MM optimization. The QM/MM model included the entire protein and 606 water molecules.

The construction of a full atomistic model of the T203I mutant was based on our S65T GFP QM/MM-optimized structure, in which Thr203 was mutated to Ile203. The mutated protein was extensively solvated and subjected to MD simulations with a protocol similar to that used for constructing the initial structure of S65T GFP. The simulations were carried out for 7 ns in the NPT ensemble at 300 K and 1 bar. The longer trajectory was used for enhanced sampling of the configurational space of the mutated T203I protein. The NPT ensemble enabled adjusting a volume of the cell during the MD run with periodic boundary conditions. Following the MD simulations, the model system was cooled down to 1 K in steps of 1 K during 0.3 ns. The final MD geometry was then obtained by performing geometry optimization for an additional 2000 steps.

In order to explore the conformational flexibility of EGFP T203I and its impact on the TPA cross-section, we used 21 different frames, which were taken from the MD trajectory in the time window from 1.5 to 6.5 ns with a step of 0.25 ns (before the annealing procedure). The structure obtained after annealing was also used in the calculations of the TPA cross-section. The MD structures were reduced in size down to 5415 atoms by removing the outer-shell water molecules. Then, the QM/MM optimization and further calculations of the spectroscopic parameters and TPA cross-sections were carried out for each geometry.

The MD simulations were performed with NAMD, Version 2.13 [60]. The CHARMM36 force field parameters [61] were used for the protein and water molecules, whereas the GFP chromophore parameters were based on those taken from [62].

### 3.2. Ground State QM/MM Optimization

The equilibrium geometry parameters of the model systems, the S65T GFP protein and its mutated T203I version, were obtained using the QM/MM approach. The QM part consisted of the GFP chromophore anion, which was based on the entire Thr65, Tyr66, and Gly67 residues and its nearest environment, including the side chains of Thr203 or Ile203, His148, Arg96, Glu222, Ser205, and water molecules, which form an extensive hydrogen bonding network around the chromophore (see Figure 1). The chromophore was here represented by an extended model with 70 atoms, which included the backbone and side chains of the residues that were covalently bonded to the conjugated system. In total, 154 and 159 atoms, including 15 capping hydrogen atoms, were assigned to this subsystem for S65T GFP and its T203I variant, respectively. Note that the GFP chromophore was in its charged state, while Glu222 was set to be protonated (neutral) in both the S65T GFP protein and its T203I mutant. The total QM charge equaled zero.

The QM part was described at the PBE0/(aug)-cc-pVDZ level of theory using 1550 (S65T GFP) and 1585 (T203I variant) basis functions. Diffuse functions were only affixed to the oxygen atoms of the chromophore. The MM part was treated with the CHARMM force field parameters. The ground state optimization was carried out by using a mechanical embedding QM/MM scheme, which implies that geometrical constraints were imposed on the QM part by treating the QM/MM boundary with classical force fields. The current choice of the QM part accounted for all the interactions, such as charge transfer, hydrogen bonding, and short-range repulsion, of the chromophore with the nearby residues. Additional constraints were imposed on the outer part of the protein located more than 7 Å from the chromophore, which was held fixed during the QM/MM optimization procedure.

The calculations were carried out with Firefly, Version 8.2.0 [63], which is partially based on the GAMESS (US) [64] source code. The QM/MM functionality in Firefly is based on the integrated version of the Tinker Molecular Mechanics Package, Version 3.7 [65].

### 3.3. Excited State QM/MM Calculations

The excited state QM/MM calculations were carried out using a reduced QM part to make the calculations slightly more feasible. The QM part included an extended model of the chromophore and all nearby residues that were directly H-bonded to its conjugated system. Overall, it included 110 and 115 atoms in the case of S65T GFP and its T203I variant, respectively. The rest of the protein was represented by a set of MM point charges, and the electrostatic field created by the MM part was incorporated into the one-electron part of the electronic Hamiltonian by means of the effective fragment potential (EFP) method [66,67]. Such an electrostatic embedding scheme enabled the QM electron density both in the ground and excited states to be polarized in the field of the protein. Note that all other EFP terms, such as the fragment (MM part) polarization, were not included in the calculations.

Vertical excitation energies were calculated using the model-space-invariant multiconfiguration quasidegenerate perturbation theory, XMCQDPT2 [68]. The (aug)-cc-pVDZ basis set was employed, with diffuse functions being only affixed to the oxygen atoms of the phenolate and imidazolinone moieties of the chromophore. The reference wave functions of the ground and excited states were constructed using the complete active space self-consistent field (CASSCF) method. The active space included all π-type valence orbitals of the GFP chromophore, except for one orbital located on the nitrogen atom of the imidazolinone ring, with 14 electrons being distributed over 13 orbitals. A state-averaging (SA) procedure was applied. Seven lowest-lying singlet CASSCF states were included in the SA procedure. The zero-order CASSCF wave functions were allowed to interact through the XMCQDPT2 effective Hamiltonian. The energies of the perturbed states were obtained as the eigenvalues of the effective Hamiltonian, while the projections of the perturbed states onto the zero-order states were defined by the corresponding eigenvectors. The XMCQDPT2 effective Hamiltonian was constructed in the frame of the model space spanned by seven CASCI zero-order wavefunctions. The energies of all semi-canonical orbitals used in the perturbation theory were defined by the diagonalization of the corresponding blocks of the DFT-based effective Fock operator in the basis of the computed CASSCF molecular orbitals. The hybrid PBE0 functional was used to construct the DFT-based Fock matrix. This was performed in order to improve the description of the diradical open-shell states and their excitation energies.

Differences in the permanent dipole moments of the ground ($S_{0}$) and target excited ($S_{1}$) states, as well as the transition dipole moments between all states were calculated at the zeroth-order of the XMCQDPT2[7]/SA(7)-CASSCF(14,13)/(aug)-cc-pVDZ//EFP theory by using the perturbed CASSCF states, which were obtained following the diagonalization of the effective Hamiltonian. The Firefly package, Version 8.2.0 [63], was used for the excited state QM/EFP calculations.

### 3.4. Gas-Phase Calculations

The MP2/(aug)-cc-pVTZ level of theory, with the cc-pVTZ basis set being augmented by a diffuse spdf shell on all oxygen atoms, was employed for optimizing the geometry parameters of the isolated HBDI anion in the ground electronic state. The vertical $\pi \pi^{*}$ excitation energies of HBDI were calculated at the XMCQDPT2[7]/SA(7)-CASSCF(16,14)/(aug)-cc-pVTZ level of theory within the active space, which comprises all valence π-type orbitals. The XMCQDPT2 effective Hamiltonian was constructed in the frame of the model space spanned by seven SA(7)-CASSCF wavefunctions. The properties were calculated at the zeroth-order of the XMCQDPT2 theory.

### 3.5. Calculation of the TPA Cross-Section

In the case of the two photons with the same energy ℏω, the macroscopic TPA cross-section is defined by [34,69]:(3)$$\sigma_{TPA}\left(\omega \right)=\frac{4\pi^{3}a_{0}^{5}\alpha \omega^{2}}{c}\;\langle \;\delta_{TPA}\rangle \;g\left(E_{n}-E_{0};\;2\omega ;\;\Gamma \right),$$
where $\langle \;\delta_{TPA}\rangle$ is the rotationally averaged TPA strength [70] in atomic units; in arbitrary units, this reads:(4)$$\langle \;\delta_{TPA}\rangle =\frac{F}{30}\delta_{F}+\frac{G}{30}\delta_{G}+\frac{H}{30}\delta_{H}.$$
*F*, *G*, and *H* are the universal coefficients, which depend on the polarization of the two photons. In the case of identical linearly polarized photons, F=G=H=2, and
(5)$$\delta_{F}=\sum_{\alpha ,\beta }S_{n0}^{\alpha \alpha }\;S_{n0}^{\beta \beta *};\;\delta_{G}=\sum_{\alpha ,\beta }S_{n0}^{\alpha \beta }\;S_{n0}^{\alpha \beta *};\;\delta_{H}=\sum_{\alpha ,\beta }S_{n0}^{\alpha \beta }\;S_{n0}^{\beta \alpha *}$$
(6)$$S_{n0}^{\alpha \beta }=\sum_{i}\frac{\mu_{ni}^{\alpha }\mu_{i0}^{\beta }+\mu_{ni}^{\beta }\mu_{i0}^{\alpha }}{E_{i}-E_{0}-\hbar \omega },$$
where $S_{n0}^{\alpha \beta }$ is the TPA transition matrix elements (the Cartesian components of the TPA tensor) in the sum-over-state formalism. $\mu_{pq}^{\gamma }$ is the Cartesian components of the electric dipole transition moments:(7)$$\mu_{pq}^{\gamma }=\langle \;\psi_{p}|\mu^{\gamma }|\psi_{q}\rangle .$$ Indices 0, *n*, and *i* stand for the initial state $S_{0}$, the final state $S_{n}$, and the intermediate states $S_{i}$ of the chromophore, with the energies $E_{0}$, $E_{n}$, and $E_{i}$, respectively. In Equation (Equation 3), $a_{0}$ is the Bohr radius, α is the fine structure constant, *c* is the speed of light, and $g(E_{n}-E_{0};\;2\omega ;\;\Gamma )$ is the phenomenological line shape function (e.g., Lorentzian, Gaussian, etc.) representing line-broadening effects, with the damping parameter Γ being usually set to 0.1 eV in most of the theoretical calculations; see, however, [34,69] for a more detailed discussion. The commonly used units for the TPA cross-section are GM (in honor of M. Göppert-Mayer); $1GM=10^{-50}\;{\mathrm{cm}}^{4}\;s/\mathrm{photon}$. Here, we used a Lorentzian line shape function with an HWHM of 0.1 eV for the conversion to GM units at resonant conditions ($E_{n}-E_{0}=2\hbar \omega$) [34].

#### Two-Level Model and N-Level Models

It has long been recognized (see, e.g., [39]) that direct calculation of the TPA cross-section through the sum-over-states formalism in Equation (Equation 6) requires that the summation over the intermediate states *i* should include both the initial and final states:(8)$$S_{n0}^{\alpha \beta }=\frac{\mu_{n0}^{\alpha }\mu_{00}^{\beta }+\mu_{n0}^{\beta }\mu_{00}^{\alpha }}{-\hbar \omega }+\frac{\mu_{nn}^{\alpha }\mu_{n0}^{\beta }+\mu_{nn}^{\beta }\mu_{n0}^{\alpha }}{E_{n}-E_{0}-\hbar \omega }+\sum_{i\neq 0,n}\frac{\mu_{ni}^{\alpha }\mu_{i0}^{\beta }+\mu_{ni}^{\beta }\mu_{i0}^{\alpha }}{E_{i}-E_{0}-\hbar \omega }$$ Setting $E_{n}-E_{0}=2\hbar \omega$ in the second term and omitting the sum over i≠0,n, one arrives at the following simple expression, sometimes called the two-level model (TLM):(9)$$S_{n0}^{\alpha \beta }\approx \frac{\left(\mu_{nn}^{\alpha }-\mu_{00}^{\alpha }\right)\mu_{n0}^{\beta }+\left(\mu_{nn}^{\beta }-\mu_{00}^{\beta }\right)\mu_{n0}^{\alpha }}{\hbar \omega }=\frac{\Delta \mu_{n0}^{\alpha }\mu_{n0}^{\beta }+\Delta \mu_{n0}^{\beta }\mu_{n0}^{\alpha }}{\hbar \omega },$$
where $\Delta \mu_{n0}^{\gamma }$ denotes the Cartesian components of the difference of the permanent dipole moments in the final and initial states. For the two identical linearly polarized photons, this leads to the following approximate TPA strength (in atomic units):(10)$$\langle \;\delta_{TLM}\rangle =\frac{8|\Delta \mu_{n0}{|}^{2}{\left|\mu_{n0}\right|}^{2}\left(2\cos^{2}\theta +1\right)}{\omega^{2}},$$
where θ is the angle between the vectors $\Delta \mu_{n0}$ and $\mu_{n0}$, and the approximate TPA cross-section is written as follows:(11)$$\sigma_{TLM}\left(\omega \right)=C\;|\Delta \mu_{n0}{|}^{2}{\left|\mu_{n0}\right|}^{2}\left(2\cos^{2}\theta +1\right)g\left(E_{n}-E_{0};\;2\omega ;\;\Gamma \right),$$
where *C* is the combination of all the constants involved.

The main advantage of the TLM is that it provides simple and easy-to-interpret expressions for the two-photon absorptivity; see Equations (Equation 10) and (Equation 11). Obviously, the TLM cannot be considered a reasonable approximation in many situations, in particular when either $\mu_{n0}=0$ (meaning that the $S_{0}\;\to \;S_{n}$ transition is forbidden as a one-photon absorption) or $\Delta \mu_{n0}=0$. However, in some other cases, the TLM approximation can give a major contribution to the TPA cross-section, for instance for the GFP-type chromophores [16].

In the present study, we explored the applicability of the two-level model to numerical calculations of the TPA cross-sections of the S65T GFP protein and its T203I modification. We also investigated the numerical convergence of the sum-over-states formalism for a series of the *N*-level models (NLMs) with increasing *N*:(12)$$S_{n0}^{\alpha \beta }\approx \frac{\Delta \mu_{n0}^{\alpha }\mu_{n0}^{\beta }+\Delta \mu_{n0}^{\beta }\mu_{n0}^{\alpha }}{\hbar \omega }+\sum_{i\neq 0,n}^{N-2}\frac{\mu_{ni}^{\alpha }\mu_{i0}^{\beta }+\mu_{ni}^{\beta }\mu_{i0}^{\alpha }}{E_{i}-E_{0}-\hbar \omega }$$

## 4. Conclusions

In the present study, we proposed a methodology for calculating the TPA cross-sections of fluorescent proteins and their mutants by using MD conformational sampling and high-level electronic structure theory in the framework of the QM/MM approach. We showed that MD annealing might be a good choice for obtaining a representative structure of the protein’s active site, when only a mean value of the TPA cross-section is needed. To access the distribution of the TPA cross-sections and, ultimately, to account for inhomogeneous spectral broadening, an extensive MD sampling is required. The use of the XMCQDPT2 theory allows one to calculate excited state properties at a sufficiently high level, providing good estimates for TPA cross-sections in a series of fluorescent proteins with the same anionic chromophore, but different local environments.

We also directly validated the applicability of the TLM, demonstrating a fast convergence of the calculated TPA cross-sections for the S${}_{0}$ → S${}_{1}$ transition in the EGFP-like proteins for a series of the NLM models with increasing *N*. In this case, the applicability of the TLM model is explained by the charge transfer character of the S${}_{0}$ → S${}_{1}$ transition, which is also optically bright upon one-photon excitation. The calculated TPA cross-sections were found to be very sensitive to changes in the permanent dipole moments between the ground and excited states. Dipole moments induced by the protein environment can significantly increase the TPA cross-section of the chromophore. In the case of the anionic form of the GFP chromophore, even a single hydrogen bond increased the TPA cross-section by a factor of two. The nonlinear photophysical properties of the chromophore anions are, thus, highly tunable by modifications of the local protein environment. Such high tunability can be used for the rational design of brighter fluorescent proteins for bioimaging using two-photon laser scanning microscopy.

## Figures and Tables

**Figure 1 ijms-24-11266-f001:**
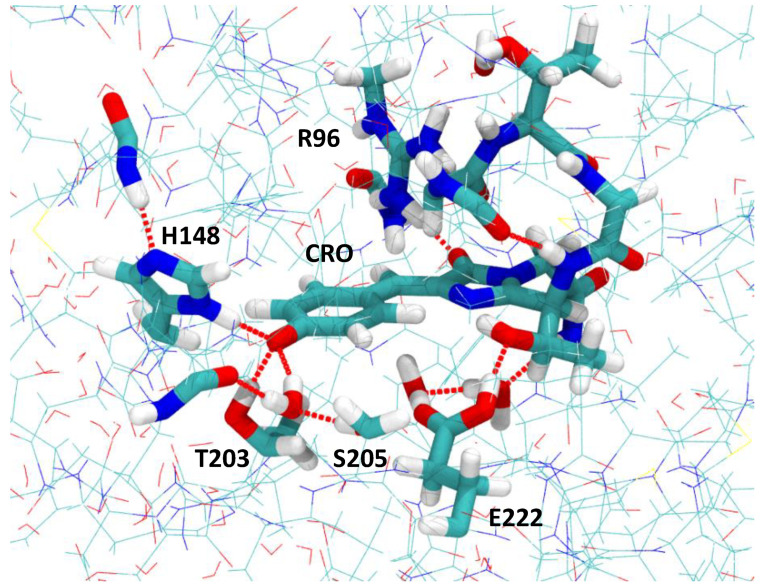
QM/MM-optimized full atomistic model of S65T GFP. The coordinates are those from Ref. [11]. Highlighted is the QM part, which consists of the chromophore (CRO) and its nearest amino acid residues involved in the hydrogen bonding network. The C, O, N, and H atoms are depicted in green, red, blue, and white, respectively. Note the disrupted hydrogen bonding network between the protonated Glu222 (E222) and the deprotonated chromophore.

**Figure 2 ijms-24-11266-f002:**
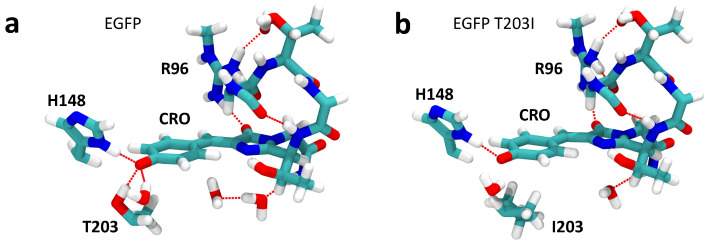
QM/MM-optimized structures of the preliminary annealed active centers of S65T GFP (**a**) and the T203I mutant (**b**), which are labeled as EGFP and EGFP T203I, respectively. The C, O, N, and H atoms are depicted in green, red, blue, and white, respectively.

**Figure 3 ijms-24-11266-f003:**
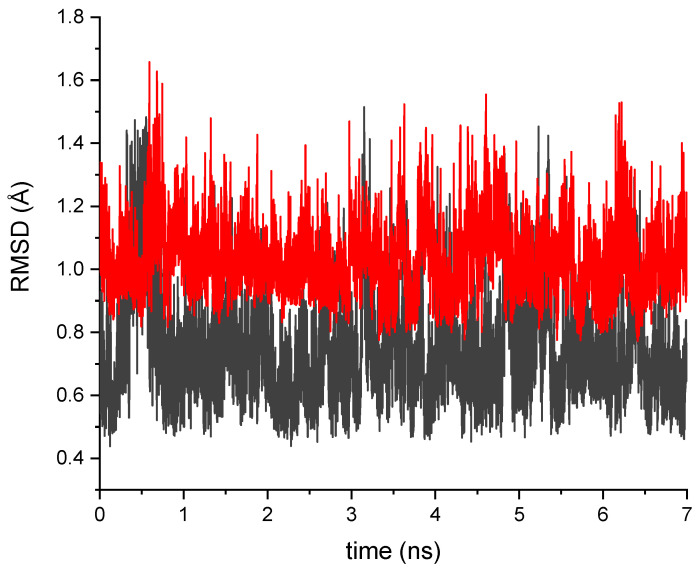
RMSD of the T203 residue along the MD trajectory for EGFP (black line) and EGFP T203I (red line) relative to the corresponding average structures.

**Figure 4 ijms-24-11266-f004:**
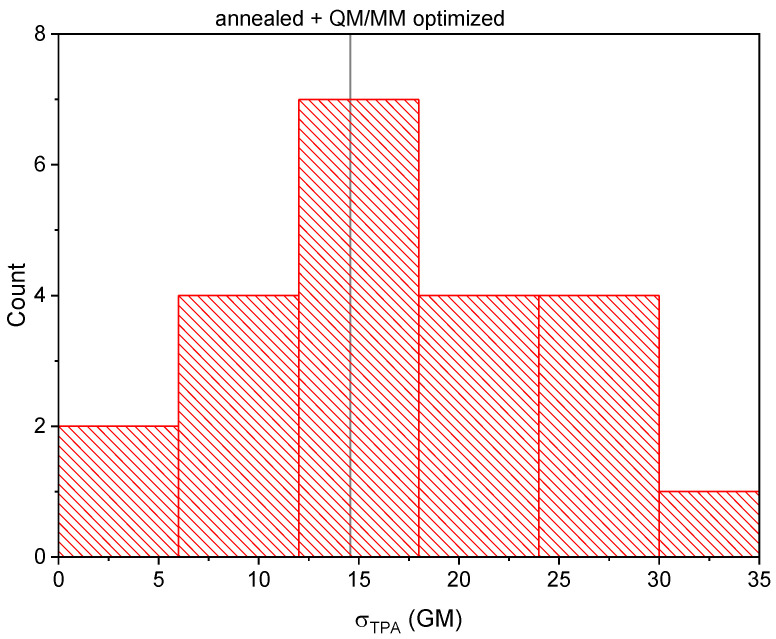
Distribution of the calculated TPA cross-sections in EGFP T203I for the structures obtained from the MD simulations followed by their QM/MM geometry optimization. The TPA cross-section of the annealed and QM/MM-optimized EGFP T203I structure is also depicted.

**Figure 5 ijms-24-11266-f005:**
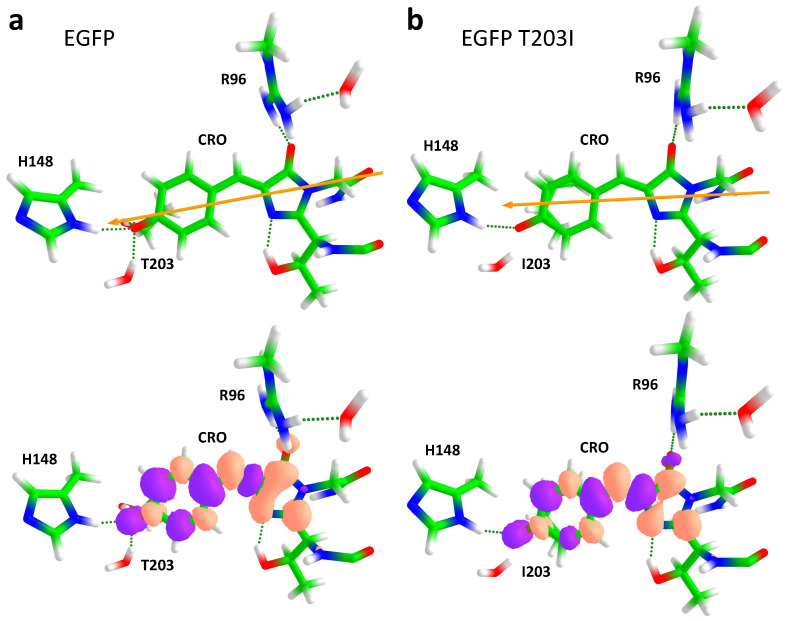
Changes in the electron density upon the S${}_{0}$→ S${}_{1}$ transition in EGFP (**a**) and EGFP T203I (**b**). The C, O, N, and H atoms are depicted in green, red, blue, and white, respectively. Upper panel: Directions of $\Delta \mu_{10}$. Note that the arrows only represent the direction, not the length of the $\Delta \mu_{10}$ vector. Lower panel: Differential electron densities. The positive and negative changes are depicted in orange and violet, respectively. Shown are the changes in the zero-order XMCQDPT2 densities.

**Table 1 ijms-24-11266-t001:** Calculated $\sigma_{TPA}$, $\delta_{TPA}$, $\mu_{10}$, $\Delta \mu_{10}$, and VEE for the S${}_{0}$→ S${}_{1}$ transition.

	HBDI	EGFP T203I	EGFP
$\mu_{10}$, D	10.5	10.9	10.5
$\Delta \mu_{10}$, D	2.9	2.5	4.4
VEE, eV	2.62	2.43	2.52
N	$\sigma_{\mathrm{TPA}},\;\mathrm{GM}\;\left(\delta_{\mathrm{TPA}},\;a.u.\right)$		
2	19 (7448)	15 (6953)	45 (19,262)
3	19 (7439)	15 (6991)	45 (19,333)
4	18 (7340)	15 (7011)	45 (19,474)
5	18 (7270)	15 (6762)	45 (19,213)
6	18 (7312)	15 (6817)	45 (19,209)
7	19 (7396)	15 (6759)	44 (19,184)
Experiment [15]	—	19	42

## Data Availability

The data are available upon proper request.

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
