# Peer review of "Impact of the Protein Environment on Two-Photon Absorption Cross-Sections of the GFP Chromophore Anion Resolved at the XMCQDPT2 Level of Theory"

_ijms, 2023, doi:10.3390/ijms241411266_

Round 1
Reviewer 1 Report
This is an interesting manuscript dealing with calculations of two-photon cross-section of the GFP chromophore anion at XMCDQDPT2 level. The theoretical approach is appropriate, but time consuming. The results were presented smoothly. Minor points should be explained: NVT and NPT ensembles were adopted at the stage of MD simulation. What's the effect if just one ensemble was adopted? Accepted as is
Author Response
Point 1: Minor points should be explained: NVT and NPT ensembles were adopted at the stage of MD simulation. What's the effect if just one ensemble was adopted?
Response 1: In this work, we did not carry out a systematic study on the influence of the ensemble selection (NVT/NPT) on the calculated TPA cross-sections. Nevertheless, the choice of the NVT ensemble for constructing a complete atomistic model of the S65T GFP protein, described in Ref. [11], was justified by the fact that this protein has a reliable X-ray structure [52]. In contrast, for the T203I mutant, the atomistic model had to be constructed manually; therefore, the NPT ensemble was used for this purpose, which made it possible to adjust a volume of the cell during the longer-timescale MD run with the periodic boundary conditions.
Reviewer 2 Report
Bochenkova and co-workers use hybrid QM/MM models to study the effect of the protein environment on the two-photon absorption (TPA) cross section of the GFP chromophore. The calculations are carried out at a multi-reference level of theory. The TPA cross sections are computed using a simple two-level sum over states model that appears to be justified here. There is a nice explanation at the end for why blue-shifted EGFP-like proteins have higher TPA cross sections, which seems to agree well with experiments.
The paper is well written, and methods for the most part are described in adequate detail. I believe this work would be interesting to the readership of IJMS. I have a few comments, mostly asking for clarification.
1. The authors mention using electronic embedding with the effective fragment potential (EFP) approach. However, what they describe in the text appears to resemble more electrostatic embedding (where the protein fixed point charges are included in the chromophore one-electron hamiltonian). I believe that the EFP approach is usually used to discuss a QM/MM treatment that includes additional interaction terms, including the polarization of the MM region. Is this indeed what is used here? If yes, I think more details need to be included in the methods section about how the protein fragments are treated. If no, I think it would be more accurate to use “electrostatic embedding” here, unless I’m missing something.
2. The GFP chromophore (HBDI) has been used as a model system for computing TPA cross sections in several studies, especially in theory / benchmarking papers related to TPA. This includes some QM/MM calculations (e.g., 10.1039/C2CP23537D) and calculations that calculations that include the first shell of amino acids surrounding the chromophore (e.g., 10.1021/acs.jpca.2c02395). There are also several other gas-phase HBDI calculations. Several of those papers have not been cited here. It would be good to cite them and maybe even add some discussion comparing the results of calculations in this work with existing computational work on HBDI’s TPA, where relevant.
3. This sentence is unclear (pg 4, line 162): “Note that the GFP chromophore is in its charged state, while Glu222 is protonated and neutral in both cases. The total QM charge equals zero.” What do the authors mean by “protonated and neutral in both cases”?
4. It appears that the T203I mutant has both the neutral and anionic form of the chromophore. However, the computations were only done for the anionic chromophore. Is this because the contribution of the neutral chromophore is expected to be minimal?
There are only minor typos (e.g., recenlty instead of recently, pg 11)
Author Response
Point 1: The authors mention using electronic embedding with the effective fragment potential (EFP) approach. However, what they describe in the text appears to resemble more electrostatic embedding (where the protein fixed point charges are included in the chromophore one-electron hamiltonian). I believe that the EFP approach is usually used to discuss a QM/MM treatment that includes additional interaction terms, including the polarization of the MM region. Is this indeed what is used here? If yes, I think more details need to be included in the methods section about how the protein fragments are treated. If no, I think it would be more accurate to use “electrostatic embedding” here, unless I’m missing something.
Response 1: In this work, an electrostatic embedding scheme was used. It was incorporated by the means of the EFP functionality within the Firefly computational package. No extra EFP terms were added. We now use "electrostatic embedding" and we have added the following sentence: "Note that all other EFP terms, such as the fragment (MM) polarization, are not included in the calculations."
Point 2: The GFP chromophore (HBDI) has been used as a model system for computing TPA cross sections in several studies, especially in theory / benchmarking papers related to TPA. This includes some QM/MM calculations (e.g., 10.1039/C2CP23537D) and calculations that calculations that include the first shell of amino acids surrounding the chromophore (e.g., 10.1021/acs.jpca.2c02395). There are also several other gas-phase HBDI calculations. Several of those papers have not been cited here. It would be good to cite them and maybe even add some discussion comparing the results of calculations in this work with existing computational work on HBDI’s TPA, where relevant.
Response 2: We have added the references in which the TPA strengths of FPs and their chromophores have been explored theoretically [Nanda2015,Beerepoot2015,Steindal2012, Ma2014, de Wergifosse2022]. We note that in most of these studies, the TDDFT quadratic response theory has been used, which provides significantly underestimated values of the TPA strengths due to the deficiency in describing the excited-state properties. The references have been added to the introduction and the following sentences have added to the discussion: "By comparing our results on the HBDI anion with those obtained previously using the TDDFT quadratic response theory [Ma2014], we conclude that the latter significantly underestimates the absolute values of the TPA strengths due to the deficiency in describing the excited-state properties. This is in agreement with previous benchmark calculations on the performance of various exchange-correlation functionals for predicting TPA strengths [Beerepoot2015,Beerepoot2018]."
Point 3: This sentence is unclear (pg 4, line 162): “Note that the GFP chromophore is in its charged state, while Glu222 is protonated and neutral in both cases. The total QM charge equals zero.” What do the authors mean by “protonated and neutral in both cases”?
Response 3: We have rephrased this sentence. Now it reads: "... while Glu222 is set to be protonated (neutral) in both the S65T GFP protein and its T203I mutant"
Point 4: It appears that the T203I mutant has both the neutral and anionic form of the chromophore. However, the computations were only done for the anionic chromophore. Is this because the contribution of the neutral chromophore is expected to be minimal?
Response 4: In contrast to the T203I mutant, where both the anionic and neutral forms of the chromophore co-exist, the chromophore in EGFP is predominantly in its anionic form; accordingly, it is the GFP chromophore anion that has been chosen to study the effect of this mutation on TPA of EGFP.
Reviewer 3 Report
The manuscript Aslopovsky et al provide a computational model for calculation the 2 photon cross section of EGFP and how it is influenced by the mutation T203I. GFP is often used for two photon fluorescence microscopy and it is important to be able to evaluate the different structural parameter that influence light absorption and fluorescence efficiency. The work here described is well done and rigorous. The methods are well explained allowing to transpose to other mutations and could be extended to other GFP like fluorescence proteins. I have a minor issue. The authors do not explain well why they chose to study this specific mutation. Aequorea wild type GFP exhibit two excitation peaks (400 nm and 470nm). The T203I preserve only the400nm. May be the authors could cite Ehrig et al FEBS letters 1995. Especially they did a great work describing the S65T mutation of EGFP that favor the 470nm excitation peak
Author Response
Point 1: I have a minor issue. The authors do not explain well why they chose to study this specific mutation. Aequorea wild type GFP exhibit two excitation peaks (400 nm and 470nm). The T203I preserve only the400nm. May be the authors could cite Ehrig et al FEBS letters 1995. Especially they did a great work describing the S65T mutation of EGFP that favor the 470nm excitation peak.
Response 1: In our studies, we have used the S65T GFP protein with the photophysical properties similar to EGFP. These proteins feature a single excitation peak at ∼490 nm and predominantly exhibit the anionic form of the chromophore. In the T203I mutant of EGFP, both the anionic and neutral forms coexist, resulting in two excitation peaks. The two excitation peaks are indeed observed in the TPA spectra of the T203I EGFP mutant (Ref. [15]). This is in contrast to T203I GFP, where there is only one peak observed in the fluorescence excitation spectrum due to the predominant neutral form of the chromophore.
In this paper, we specifically study the effect of the protein environment on the GFP chromophore anion. We choose the T203I mutation, which is close to the chromophore, and demonstrate that in the case of the GFP chromophore anion, even a single hydrogen bond is capable of drastically increasing the TPA cross-section.
As suggested by the Referee, we have added the abovementioned reference to the main text to distinguish between the T203I GFP and T203I EGFP: "We note that in contrast to T203I EGFP, the T203I GFP protein without the key S65T mutation exhibits only a single excitation peak due to the neutral form of the chromophore [Ehrig1995]."